# Effect of Guar Gum Content on the Mechanical Properties of Laterite Soil for Subgrade Soil Application

**DOI:** 10.3390/polym16152202

**Published:** 2024-08-02

**Authors:** Shailendra Pandurang Banne, Saurabh Kulkarni, Jair Arrieta Baldovino

**Affiliations:** 1Department of Civil Engineering, Pimpri Chinchwad College of Engineering, Nigdi, Pune 411044, India; 2Department of Civil Engineering, MES’s Wadia College of Engineering, Pune 411001, India; saurabh.kulkarni@mescoepune.org; 3Civil Engineering Program, Universidad de Cartagena, Cartagena de Indias 130015, Colombia; jarrietab2@unicartagena.edu.co

**Keywords:** biopolymers, guar gum, laterite soil, soil stabilization, shear strength, CBR, pavement design, cost analysis

## Abstract

Using biopolymers for soil stabilization is favorable compared to more conventional methods because they are more environmentally friendly, cost-effective, and long-lasting. This study analyzes the physical properties of guar gum and laterite soil mixes. A comprehensive engineering study of guar gum-treated soil was conducted with the help of a brief experimental program. This study examined the effects of soil–guar gum interactions on the strengthening behavior of guar gum-treated soil mixtures using a series of laboratory tests. The treated laterite soil’s dry density increased marginally, while its optimum moisture content decreased as the guar gum increased. Treatment with guar gum significantly enhanced the strength of laterite soil mixtures. For laterite soil with 2% guar gum, the unsoaked CBR increased by 148% and the soaked CBR increased by 192.36%. The cohesiveness and internal friction angle increased by 93.33% and 31.52%, respectively. These results show that using guar gum dramatically improves the strength of laterite soil, offering a more environmentally friendly and sustainable alternative to traditional soil additives. Using guar gum in T8 subgrade soil requires a 1395 mm pavement depth and costs INR 3.83 crores, 1.52 times more than laterite soil. For T9 subgrade soil, the depth was 1495 mm, costing INR 4.42 crores, 1.72 times more than laterite soil. This study introduces a novel approach to soil stabilization by employing guar gum, a biopolymer, to enhance the physical and mechanical properties of laterite soil. Furthermore, this study provides a detailed cost–benefit analysis for pavement applications, revealing the financial feasibility of using guar gum despite it requiring a marginally higher initial investment.

## 1. Introduction

Roads, buildings, power plants, waste treatment facilities, and other structures associated with civil engineering are all part of infrastructure development. Population growth and the development of the construction industry have increased the demand for working on problematic soils. The practicability of the site is an essential consideration in the choice of project; in general, one should choose soil with a high bearing capacity. Most geotechnical projects require ground modification since it is impossible to find a construction field that matches the design criteria without it [1]. Stabilization refers to a method used to enhance the strength properties of soil by adding more material to the existing soil [2]. Soil treatment methods for geotechnical applications must be understood and implemented. Most soil improvement methods are mechanical or chemical, but a mix of both is sometimes used. Chemical treatment is a low-cost and easy-to-apply strategy for improving soil engineering properties. Chemical soil stabilizers are typically classified using traditional or nontraditional additives: cement, bituminous materials, lime, and industrial byproducts such as fly ash, granulated blast-furnace slag, and calcium carbide residue are examples of traditional additives [3]. The more common additives include fly ash, calcium carbide residue, and granulated blast-furnace slag [4]. Enzymes, resins, polymers, silicates, acids, ions, and lignin derivatives in liquid or powder form are examples of nontraditional additives [5]. Biopolymers are environmentally friendly, cost-effective, and non-toxic nontraditional additives [3]. In this study, guar gum is used to stabilize laterite soil.

Many researchers are working on stabilizing laterite soil. The influence of lime and cement on laterite soil [6,7,8] has also been studied. The optimum amount of strength was determined to be achieved when the lime content was increased to 12%: the compaction rate of the treated soil was increased to 86.2%, and the field CBR value was increased to 112.44% [9] for road bases. The properties of laterite soil have also been significantly improved using fly ash [10], bagasse ash [11], and rice husk ash (RHA) [12]. The deformation of a physical model of laterite road bases in Halmahera was tested and found to be significantly reduced at a lime concentration of 10% [13]. The time-dependent responses of laterite soil to two types of unconventional additives (TX-85 and SH-85) were investigated using macro and microstructures [14]. Lateritic soil was further treated using tire-derived aggregates and 3% micro silica [15], groundnut shell ash [16], CaCl_2_ [17], polymer soil stabilizer (SS299) [5], agricultural wastes [18], cement and quarry dust [19], lignin [20], cellulose [21] and xanthan gum, [22,23,24].

The above paragraph has discussed the different additives used in laterite soil stabilization. Incorporating these additives has shown promising improvements in the mechanical and geotechnical properties of laterite soil. By enhancing the soil’s strength, durability, and stabilitsy, these stabilization techniques contribute significantly to sustainable construction practices.

Guar gum is a natural, high-molecular-weight polysaccharide derived from the endosperm of guar beans (Cyamopsis tetragonoloba). A program of experiments was conducted to investigate the behavior of guar gum-mixed soil [25]. The results showed that a 2% inclusion of guar gum induces the most significant modification of the geotechnical properties. Soil enhancement using biopolymers is appealing since they are more environmentally friendly and long-lasting than alternatives like cement [26]. Xanthan gum and guar gum, two naturally occurring and affordable biopolymers, were examined for their ability to stabilize mine tailings (MTs) [27]. Biopolymers’ effects on kaolin clay’s mechanical behavior were also investigated: guar gum specimens increased the strength by up to 8.53 times [28]. Compared to cement and other frequently used materials, guar gum is more durable and environmentally friendly, making it a potential material for soil improvement. Based on simple cost comparisons, it is not more affordable than cement [29]. Biopolymer-treated soils considerably increase shear strength and decrease native expansive material shrinkage [30]. Guar gum significantly lowers the permeability at different densities because the particles stick together, filling the pores. This property has the potential to play an essential part in the stability of soils that are prone to collapse [31]. With the inclusion of guar gum, the engineering properties of clay are increased, making the material more flexible and valuable for a broader range of uses [32]. These insights underline the importance of sustainable and environmentally friendly solutions in geotechnical engineering. This exploration aims to contribute to the body of knowledge on biopolymer-based soil stabilization, offering a promising alternative to traditional methods and paving the way for innovative applications in sustainable infrastructure development.

In the present study, stabilized laterite soil with guar gum was utilized as subgrade soil. Previous research has extensively examined the incorporation of various additives into subgrade soil to enhance its properties. Lime [33], cement kiln dust (CKD) [34], plastic trash [35], portland cement (PC) [36], fiber-reinforced fly ash [37], coir fiber [38], fiber-reinforced ash [39], cement and lime [40], and SSA–cement [41] have all been added to test the subgrade material of flexible pavement according to strength and deformation criteria. The pavement thickness can be lowered based on the increased CBR. This study contributes to this body of knowledge by exploring the effects of guar gum stabilization on the performance characteristics of laterite soil used in subgrade applications.

Recent experiments have started studying many aspects of the behavior of laterite soil and guar gum mixes. There is also a lack of information on the shear strength, compaction parameters, and geomorphological characteristics of guar gum-stabilized laterite soils from Parshuram Ghat, Chiplun, India. This investigation used an experimental approach to analyze how adding guar gum affects fine-grained laterite soil. Parallel to this, the change in the soil microstructure was studied using SEM tests. The results of these tests help us to comprehend the efficiency of laterite soil when guar gum is used for stabilization.

## 2. Materials and Methods

### 2.1. Laterite Soil

This study used local laterite soil. The current study utilized laterite soil from Parshuram Ghat [Figure 1a], Chiplun Maharashtra, from a trial pit of 1.0 m depth from four different areas [soil types A, B, C, and D] (17°33′28.0″ N, 73°29′53.0″ E). Table 1 shows the laterite soil’s particle size distribution, specific gravity, consistency, compaction, CBR, and permeability values. The grain size investigation found 29–32% sand, 19–25% silt, and 52–58% clay in the soil. Figure 2 depicts the soil’s grain size distribution. These results classify the soil as highly compressible silt (MH) under Indian standards (IS: 1498–1970) [42]. LL was 58–64%, PL 34–39%, and PI 24–25%. In the heavy compaction test, the MDD was found to be 1.52–1.68 g/cc, and the OMC was found to be 13–16%. Soaked and unsoaked laterite soil CBR values were 4% and 6%, respectively. The laterite soil’s cohesiveness was 0.13 to 0.15 kg/cm^2^, and its internal friction was 13.09 to 15.22 degrees. Heavy compaction had a permeability coefficient of 2.75 × 10^−4^ cm/s. The properties of the laterite soil at different locations were almost identical.

#### XRD Analysis of Laterite Soil

According to the study, oxygen (O) made up the largest share, at 44.3%, followed by phosphorus (P), which made up 21.5%. Elements in the soil were mainly germanium (Ge), which made up 20.1%; sodium (Na), which made up 9.1%; and aluminum (Al), at 5.0%. These results shed light on the varied chemical makeup of the laterite soil. Figure 3 and Figure 4 show the intensity plotted against the 2ϴ value and the laterite soil’s chemical makeup from the 2ϴ values that go with it.

### 2.2. Guar Gum

An all-natural biopolymer extracted from the guar seed’s endosperm polysaccharide, guar gum is known by its scientific name, Cyamopsistetragonolobus. Guar gum comes from the guar seed, a leguminosae plant [43]. Linear 1,4-linked mannose residues form the backbone of guar gum, with galactose residues 1,6 linked to every other mannose to create small side branches that will dissolve quickly and stabilize well. It is hydrocolloidal and nonionic in water.

#### XRD Analysis of Guar Gum

Oxygen (O) content: The oxygen component in guar gum plays a crucial role in its ability to form strong hydrogen bonds with water molecules. This property enhances its water-retention capacity and aids in improving soil moisture content. By increasing soil moisture levels, guar gum helps to prevent excessive drying, which can lead to soil erosion and loss of structural integrity. Carbon (C) content: Guar gum enhances soil structure binding when added to the soil. Additionally, the carbon component in guar gum can act as a binder, helping to stabilize soil particles and prevent erosion. Phosphorus (P) content: Phosphorus is an essential nutrient for plant growth and development. Strong and healthy root systems stabilize soil by binding soil particles, preventing erosion and improving soil structure. Figure 5 and Figure 6 present the guar gum’s XRD analysis and its elemental composition.

### 2.3. Sample Preparation

The soil samples were dried in an oven for 24 h at 100–105 degrees Celsius. The powdered guar gum was mixed with the required water content to make a viscous gel at concentrations of 1%, 2%, and 3% based on the soil weight (wet mixing method) [25]. The guar gum powder was dispersed over a large tray, with water lightly sprayed onto it to prevent clumping. Water was then gradually added and mixed thoroughly to avoid lumps, ensuring that the guar gum was uniformly covered. The guar gum was mixed with water and coated. Hydrating the soil matrix with guar gum reduces evaporation. The laterite soil was treated with the gel after 24 h. The guar gum evenly coated each soil particle when the laterite soil and gel were mixed.

## 3. Results and Discussions

### 3.1. Atterberg’s Limit

Because guar gum hydrates and forms strong hydrogels via hydrogen bonding, increasing the guar gum content in treated soil increases its liquid and plastic limits (Figure 7). Additionally, the threshold for shrinkage increases for the same reason. This is because the hydrogels formed by the guar gum provide additional structure and stability to the soil, reducing its tendency to shrink. The effectiveness of this transformation depends critically on the amount of guar gum used. Guar gum’s hydrogen bonding increases hydrogel formation at higher concentrations. At higher concentrations of guar gum, the extent of hydrogen bonding increases, leading to the formation of more robust hydrogels.

The properties of the laterite soil indexed according to guar gum concentration are shown in Table 2.

### 3.2. Modified Proctor Test

Road, runway, earth dam, embankment, and earth wall construction processes are governed by soil density. Factors such as soil density impact properties such as the settlement rate, bearing capacity, shear strength, and permeability. The compaction properties of stabilized soil are affected by the soil type, treatment method, stabilizer nature, and compaction technique [25]. The dry unit weight of guar gum-treated soil depends on the soil type and the solution viscosity. This research therefore analyzed the effects of varying amounts of guar gum on soil compaction.

It has been demonstrated that the optimal moisture content (OMC) of soils that have been treated with guar gum will decrease, whereas the dry unit weight of the soil will increase as a result of this treatment, as seen for all samples (Types A, B, C, D) of laterite soil in this study. This change is insignificant in magnitude, particularly concerning the most excellent dry density possible (Figure 8). Gum solutions reduce the friction between soil matrix particles. The soil’s total dry unit weight thus increased slightly. Due to its high viscosity, the 2% guar gum treatment reduced the dry unit weight. The gum solution resists compaction. Guar gum can increase the dry unit weight by up to 2%, but then it decreases. Higher gum concentrations separate the soil particles into a highly viscous solution that coats them. The soil dry unit weight then decreases. This occurs because the solution coats the particles. Because guar gum needs water to hydrate, the optimal moisture level in mixtures of laterite soil and guar gum is reduced when water is absorbed. Guar gum usually continues the hydrogel development process until a complete viscous gel is formed due to the formation of strong polymer bonds.

### 3.3. Direct Shear Test

For this section, we used a direct shear box test to examine the behavior of both natural and artificially altered laterite soils. Figure 9 shows the highest shear stress and normal stress in four soil samples with 1%, 2%, and 3% guar gum concentrations. The comparison results are shown here. The shear box test results for the untreated and treated soils are shown in Table 3.

A stronger interlocking and bonding between soil particles was observed after introducing particles into the soil matrix. As a result, the direct shear strength increased due to the roughening of particle surfaces caused by the formation of compounds. Therefore, it may be inferred that adding up to 2% guar gum to laterite soil improves the strength parameter cohesiveness of the lateritic soil, whereas adding more guar gum decreases its cohesion.

### 3.4. CBR Test

Figure 10 shows how the CBR changed with the amount of guar gum in the four samples. In a CBR test, soil that has been stabilized is capable of being compacted to a greater density. Because of this, the stabilized soil was strengthened, and as a result, the amount of load required for penetration was reduced, which increased the CBR. In addition, the proportion of guar gum in the soil is directly related to the soil sample’s CBR value. According to the results that we recorded, a peak value of 6% was obtained by using 3% guar gum in an unsoaked condition, but this value was almost 4.2% when the gum was used in a soaked condition for all four soil samples (Types A, B, C, D). The CBR values for both the unsoaked and soaked samples increased with guar gum concentrations up to 2%, but beyond this, the guar gum content increasing caused little or no subsequent change in the CBR values.

### 3.5. Microstructural Study

#### 3.5.1. XRD Analysis of LS + 2% GG

An XRD analysis of the laterite soil with 2% guar gum (the optimum dose) was carried out. The chemical composition in the mix was observed to be oxygen (O) 48.50%, phosphorus (P) 31.30%, and calcium (Ca) 20.20%. Laterite soil and guar gum mixtures, with a high oxygen content, such as 48.50%, typically exhibit a significant presence of oxygen-containing compounds like silicates, alumino-silicates, and various oxides. The high oxygen content suggests a strong potential for hydrogen bonding, which can influence the soil’s physical and chemical properties. These hydrogen bonds enhance the soil’s ability to retain moisture, impact its plasticity, and contribute to the stability of organic and inorganic complexes within the soil matrix. Guar gum was mixed with laterite soil, and then various engineering properties of the soil improved. Guar gum can improve soil structure and reduce erosion by binding soil particles together, increasing their cohesion. This is particularly useful in areas prone to erosion, such as slopes or construction sites. Guar gum can enhance the soil’s physical properties, such as its porosity, permeability, and compaction. It can help stabilize loose or sandy soils, making them more suitable for construction activities. Figure 11 and Figure 12 depict the XRD results for the laterite soil and 2% GG mixture and its elemental composition.

#### 3.5.2. SEM (LS + 2% GG)

Scanning electron microscopy (SEM) micrographs were used to learn more about improving the shearing parameters of laterite soil and guar gum mixtures. The micrographs showed an accumulation of guar gum inside the soil gaps, which not only helps to create hydrogel but also helps to join the individual soil particles together. The strength of soil treated with guar gum depends on how strong and dense these links are in the pores. The creation of hydrogels modifies the character of the soil matrix when it is stabilized with guar gum. Figure 13 shows the SEM micrographs of the laterite soil mixture with 2% guar gum. The hydrogel cementitious products are highlighted in marked circles.

## 4. Pavement Design and Cost Analysis

Pavement design and cost analysis are crucial aspects of road and highway projects. Pavement design involves selecting appropriate materials and configuring layers for roads, highways, parking lots, and other transportation surfaces to withstand the expected traffic loads and environmental conditions. On the other hand, cost analysis involves evaluating the financial aspects of pavement construction and maintenance to ensure that the chosen design is cost-effective over its lifecycle. Cost analysis in pavement design consists of assessing the financial implications of various design alternatives over the pavement’s lifecycle. This includes both initial construction costs and maintenance expenses. The design approach for these pavements adheres to the guidelines outlined in IRC: SP: 72–2015 [44]. The intricate nature of these pavements demands a more robust design strategy, as outlined in IRC: 37-2018 [45]. In the current investigation, the suggested low-volume pavement design follows IRC: SP: 72-2015 [44]. The traffic, quantified according to the cumulative Equivalent Single-Axle Load (ESAL), was categorized into nine groups (T1 to T9). The focus of the examination was directed towards two specific traffic categories, namely T8 and T9, each of which pertains to distinct ranges of Equivalent Single-Axle Loads (ESALs). The ESAL values, a metric employed to quantify the cumulative impact of axle loads on a pavement’s structural integrity, ranged from 1,000,000 to 1,500,000 for T8 and 1,500,000 to 2,000,000 for T9. By explicitly considering T8 and T9, this analysis aims to capture the influence of heavier traffic intensities on the pavement’s performance.

### 4.1. Guar Gum Used in Solely Subgrade Soil vs. in Subgrade Soil with Embankment Soil (T8 Category)

When utilizing guar gum in subgrade soil within the T8 category, the complete depth necessary for the pavement amounted to 1395 mm, incurring a cost of INR 3.83 Crores. This expense was 1.52 times higher in comparison to using only laterite soil. Similarly, applying guar gum in subgrade soil alongside embankment soil, requiring a total thickness of 1420 mm, incurred a cost of INR 5.48 Crores. This represented a cost increase of 2.17 times when compared to using solely laterite soil. In contrast to xanthan gum, guar gum necessitated a lower cost outlay. The optimal proportion for guar gum was 2% relative to the dry weight of the laterite soil. Table 4 and Table 5 show the pavement design and cost analysis for the guar gum case under the T8 category.

### 4.2. Guar Gum Used in Solely Subgrade Soil vs. in Subgrade Soil with Embankment Soil (T9 Category)

When employing guar gum within subgrade soil classified under the T9 category, the total required pavement depth was 1495 mm, resulting in an expenditure of INR 4.42 Crores. This cost was 1.72 times higher compared to using only laterite soil. Similarly, using guar gum in subgrade soil combined with embankment soil, necessitating an overall thickness of 1420 mm, led to a cost of INR 5.86 Crores. This presented a cost escalation of 2.28 times relative to the exclusive use of laterite soil. In contradistinction to xanthan gum, implementing guar gum demanded a more economical investment. The optimal ratio for guar gum was determined to be 2% in relation to the dry weight of the laterite soil. Table 6 and Table 7 show the pavement design and cost analysis of utilizing guar gum biopolymer within the T9 category.

## 5. Conclusions

Biopolymers like guar gum, found naturally and in large amounts, are crucial for developing environmentally friendly ways to stabilize soil. Guar gum, depending on the type of biopolymer, its concentration, and the type of soil, significantly affects soil’s physical and mechanical properties. According to the results, adding guar gum at a dosage of 2% results in the most significant improvement in the observed geotechnical properties. This study’s findings lead to the following conclusions:When the soil was treated with guar gum, increases of 36.03% in the liquid limit, 49.21% in the plastic limit, and 76.42% in the shrinkage rate were found.Guar gum increased the maximum dry density and decreased the optimum moisture content. The addition of 2% guar gum increased the dry unit weight of laterite soil from 1.42 to 1.81 gm/cc, but further additions lowered it.The unsoaked CBR value of soil containing 2% guar gum indicated an increase of 148%, while the soaked CBR value increased by 192.36%; further additions of guar gum did not affect the CBR values.The cohesiveness of the soil admixed with 2% guar gum increased by 93.33%, and the angle of internal friction increased by 31.52%; however, no further rise in its cohesion or angle of internal friction was found. Soils treated with guar gum improved in their shear strength.Guar gum is a promising substance for soil enhancement since it is more sustainable and environmentally friendly than other additives. However, it is currently not cost-effective. As increased production of guar gum will lower its prices, using biopolymers as soil improvement materials may become economically viable due to this research.Considering both the cost factor and the substantial enhancements observed in the soil’s mechanical properties, this study recommends guar gum as a well-suited material for integration into subgrade laterite soil. Using guar gum in T8 subgrade soil necessitated a pavement depth of 1395 mm, entailing a cost of INR 3.83 Crores, 1.52 times higher than that for laterite soil. For T9 subgrade soil, the required depth was 1495 mm, with expenses amounting to INR 4.42 Crores, representing a 1.72-times increase compared to that of laterite soil.

## Figures and Tables

**Figure 1 polymers-16-02202-f001:**
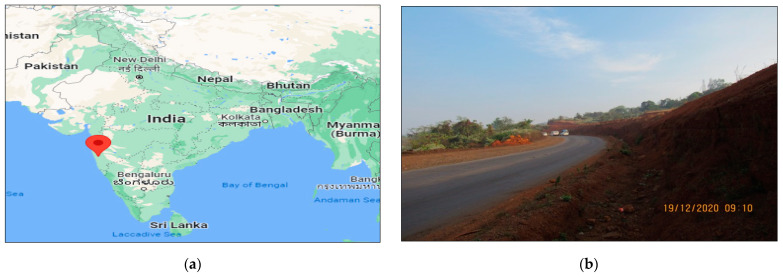
(**a**) Soil sample collection map [https://www.google.com/maps/search/lote+parshuram+ghat/@17.5645364,73.4541636, accessed on 20 November 2020]. (**b**) Soil collection site (17°33′28.0″ N, 73°29′53.0″ E).

**Figure 2 polymers-16-02202-f002:**
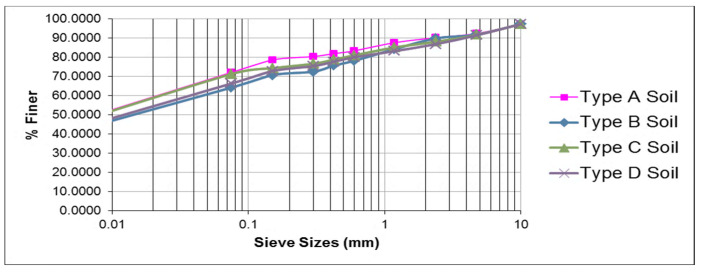
The curve of particle size distribution for the laterite soil.

**Figure 3 polymers-16-02202-f003:**
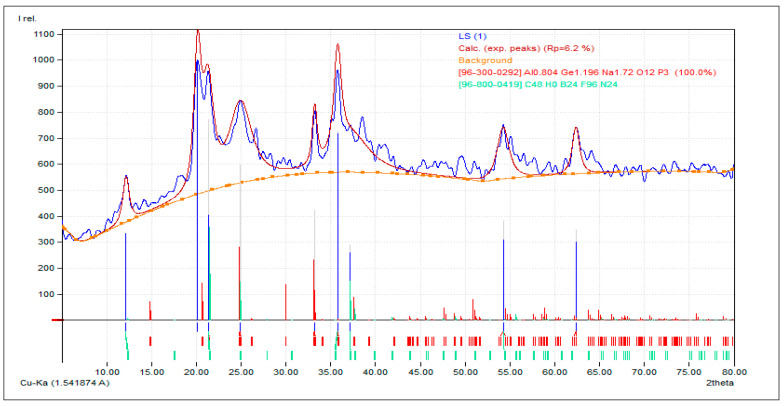
Intensity vs. 2ϴ curve for the XRD analysis of the laterite soil.

**Figure 4 polymers-16-02202-f004:**
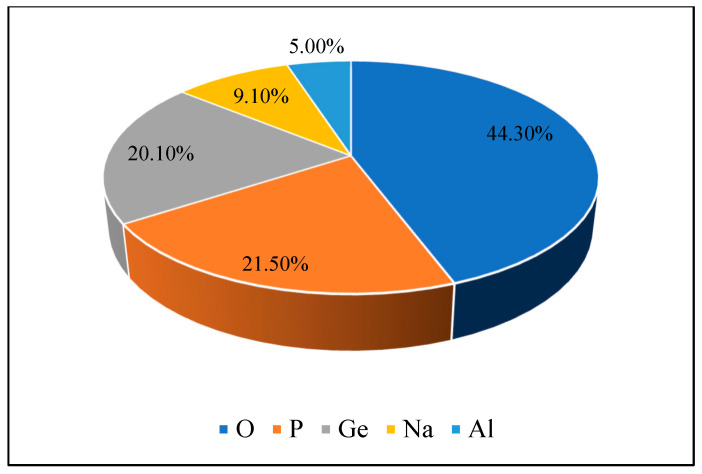
Elemental composition of the laterite soil.

**Figure 5 polymers-16-02202-f005:**
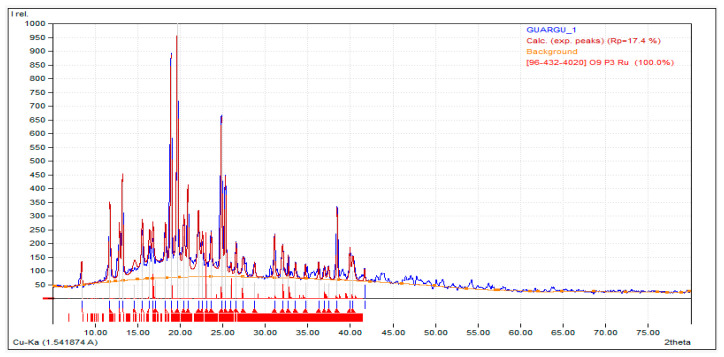
Intensity vs. 2ϴ curve for the XRD analysis of the guar gum.

**Figure 6 polymers-16-02202-f006:**
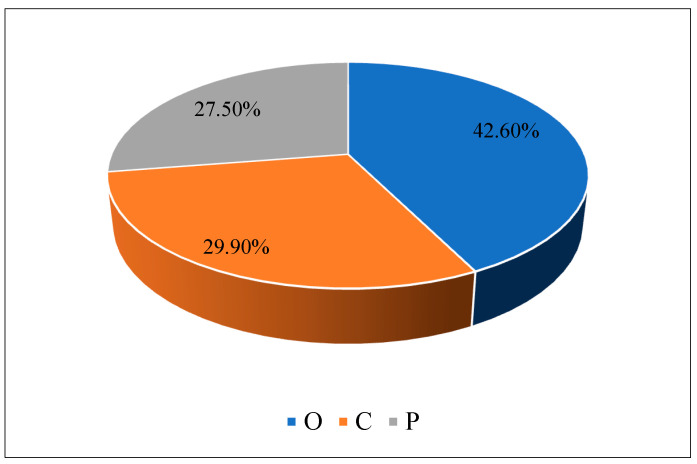
Elemental composition of the guar gum.

**Figure 7 polymers-16-02202-f007:**
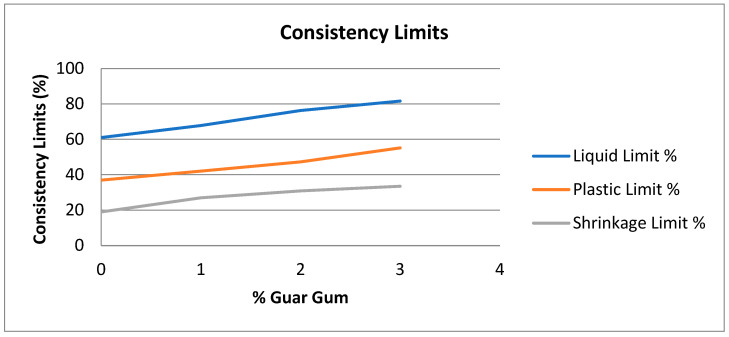
Consistency limits of the laterite soil–guar gum mixture.

**Figure 8 polymers-16-02202-f008:**
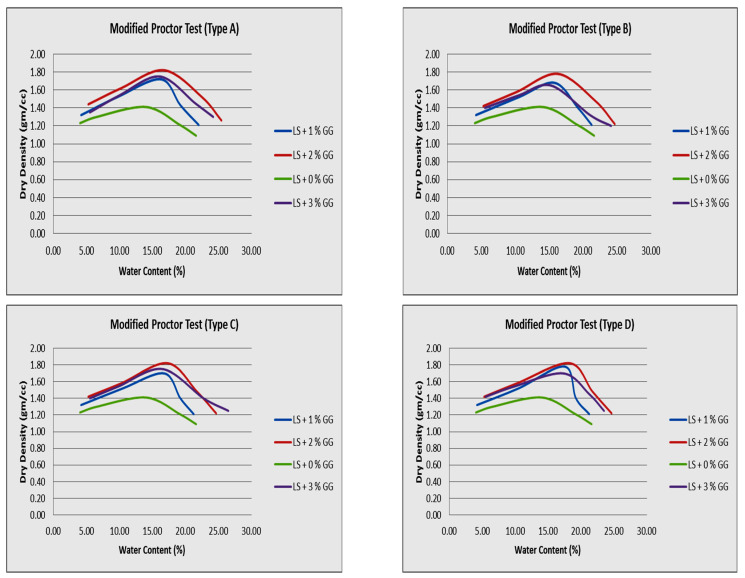
OMC and MDD values for soil samples with different percentages of guar gum.

**Figure 9 polymers-16-02202-f009:**
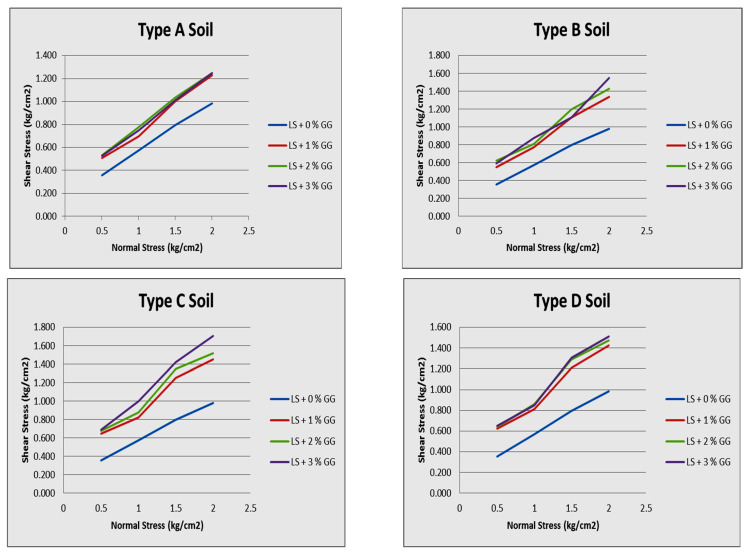
Direct shear test results for different samples (A, B, C, and D) of laterite soil.

**Figure 10 polymers-16-02202-f010:**
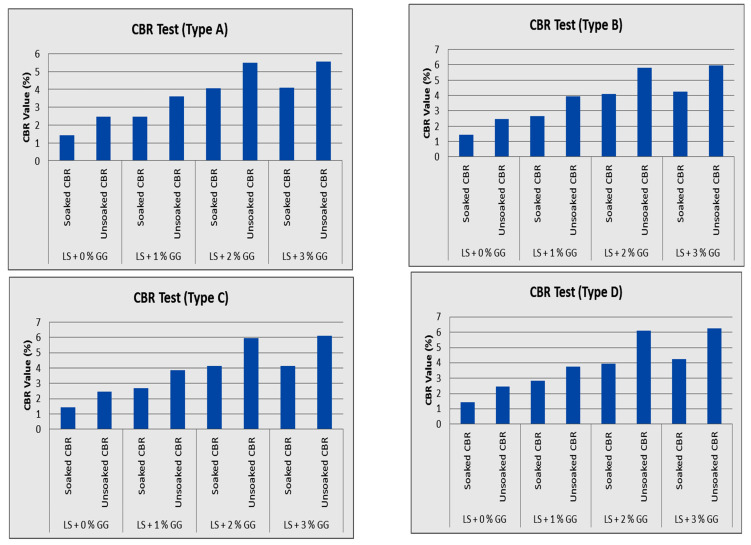
CBR test results for different samples of laterite soil.

**Figure 11 polymers-16-02202-f011:**
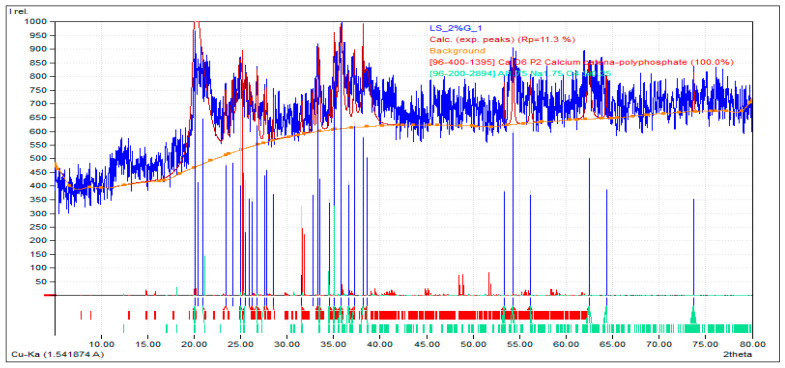
Intensity vs. 2ϴ curve for the XRD analysis of the LS + 2% GG soil mix.

**Figure 12 polymers-16-02202-f012:**
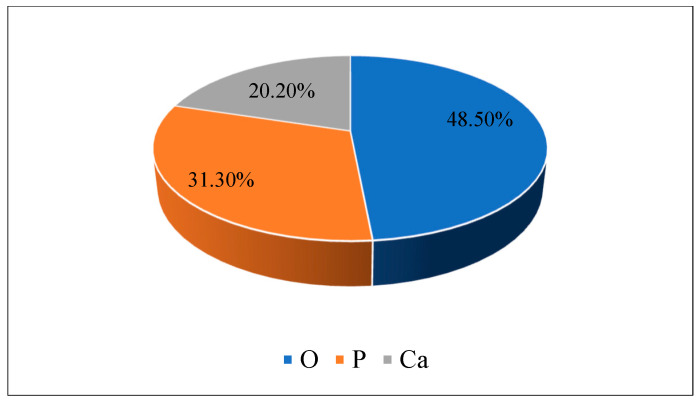
Elemental composition of the LS + 2% GG soil mix.

**Figure 13 polymers-16-02202-f013:**
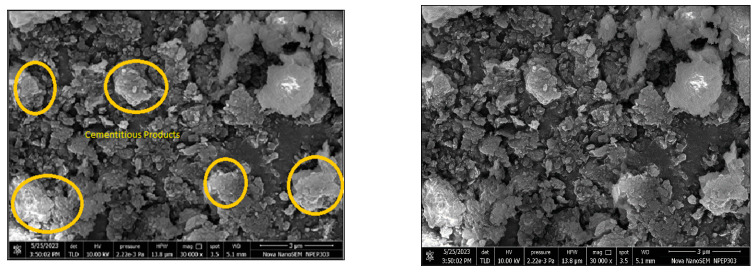
Micrographs of LS + 2% GG.

**Table 1 polymers-16-02202-t001:** Field-collected laterite soil properties.

Sr. No.	Properties	Value	Reference
1	Specific Gravity (G)	2.52–2.67	IS 2720 (Part-3) 1980 [42]
2	Distribution of Particle Size		IS: 2720 (Part-4): 1985 [42]
	(a) Sand %	29–32	
	(b) Silt %	19–25	
	(c) Clay %	52–58	
3	Consistency Limits		IS 2720 (Part-5) 1985 [42]
	LL (%)	58–64	
	PL (%)	34–39	
	PI (%)	24–25	
	SL (%)	17–21	
4	Soil Classification	MH	
5	Modified Proctor Test		IS 2720 (Part-8) 1980 [42]
	(a) Max Dry Density (gm/cc)	1.40–1.43	
	(b) OMC (%)	13.02–16	
6	CBR Value (%)		IS 2720 (Part-16) 1987 [42]
	Soaked	1.34–1.44	
	Unsoaked	2.3–2.5	
7	Direct Shear Test		IS 2720 (Part-15) 1986 [42]
	Cohesion (kg/cm^2^)	0.13–0.15	
	Angle of Internal Friction (Degree)	13.09–15.22	
8	Coefficient of Permeability (cm/sec)	2.75 × 10^−4^	IS 2720 (Part-17) 1986 [42]

**Table 2 polymers-16-02202-t002:** Laterite soil properties with varying amounts of guar gum.

Geotechnical Property	Type A Soil			Type B Soil			Type C Soil			Type D Soil			Mean		
Guar Gum %	1	2	3	1	2	3	1	2	3	1	2	3	1	2	3
Specific Gravity	2.73	2.80	2.84	2.65	2.74	2.81	2.68	2.76	2.84	2.72	2.78	2.86	2.69	2.77	2.83
Liquid Limit %	69	77	82.5	67	74	79	68	78	83	67	76	82	67.75	76.25	81.62
Plastic Limit %	40.83	48.97	56.80	41.25	46.55	54.35	40.25	47.65	53.60	42.10	45.75	56.10	42.02	47.23	55.21
Plasticity Index %	28.17	28.03	26.5	25.75	27.45	24.65	27.75	30.35	29.4	24.9	30.25	25.9	25.73	29.02	26.41
Shrinkage Limit %	26.81	29.41	32.20	25.33	31.25	34.55	28.30	32.40	33.10	27.35	30.45	34.25	26.95	30.87	33.52
MDD (kN/m^3^)	1.72	1.82	1.75	1.68	1.78	1.65	1.70	1.82	1.74	1.78	1.82	1.70	1.72	1.81	1.71
OMC %	16.25	16.75	16.10	15.75	16.25	15.25	16.75	17.25	16.5	17.5	18.25	17.40	16.56	17.12	16.31

**Table 3 polymers-16-02202-t003:** Results of the direct shear test.

Soil Description	Type A Soil		Type B Soil		Type C Soil		Type D Soil	
	C (kg/cm^2^)	Φ (Degree)	C (kg/cm^2^)	Φ (Degree)	C (kg/cm^2^)	Φ (Degree)	C (kg/cm^2^)	Φ (Degree)
Laterite Soil (LS + 0% GG)	0.15	22.81	0.15	22.81	0.15	22.81	0.15	22.81
Lateritic Soil + 1% Guar Gum (LS + 1% GG)	0.24	26.35	0.27	28.23	0.33	29.46	0.31	29.24
Lateritic Soil + 2% Guar Gum (LS + 2% GG)	0.29	25.71	0.31	29.29	0.35	31.04	0.33	30.41
Lateritic Soil + 3% Guar Gum (LS + 3% GG)	0.27	25.96	0.25	31.92	0.33	34.87	0.32	31.29

**Table 4 polymers-16-02202-t004:** Pavement design and cost analysis for guar gum used in only subgrade soil (T8 category).

Case 1: 2% Guar Gum in Only Subgrade Soil								
Length of road (m)		1000		MDD (kg/m^3^)		1820		
Carriageway width (m)		14		Biopolymer		Guar gum		
Effective CBR (%)		2.90		Layer to be used		Subgrade		
Subgrade class		S2		Dosage		2%		
Layer	Thickness (mm)	Top width (m)	Bottom width (m)	Length (m)	Qty. (cum.)	Qty. (kg)	Rate (INR)	Amount (INR)
OGPC	20	14	14	1000	280	0	267	74,760
WBM Grade-3	75	14	14	1000	1050	0	2180	2,289,000
WBM (CBR > 100%)	150	14	14	1000	2100	0	2180	4,578,000
Granular sub-base (CBR > 20%)	150	14.98	15.58	1000	2292	0	2039	4,673,388
Modified subgrade	200	15.58	16.38	1000	3196	0	3627	11,591,892
Subgrade	300	15.58	16.78	1000	4854	0	393	1,907,622
Embankment	500	16.78	18.78	1000	8890	0	393	3,493,770
Total	1395	-	-	-	-	-	-	-
Guar gum	-	-	-	-	176,686		55	9,717,708
Total								38,326,140
Total in Cr.								3.83

**Table 5 polymers-16-02202-t005:** Pavement design and cost analysis for guar gum used in subgrade soil and embankment soil (T8 category).

Case 2: 2% Guar Gum in Subgrade and Embankment Soil								
Length of road (m)		1000		MDD (kg/m^3^)		1820		
Carriageway width (m)		14		Biopolymer		Guar gum		
Effective CBR (%)		12.57		Layer to be used		Subgrade and embankment		
Subgrade class		S5		Dosage		2%		
Layer	Thickness (mm)	Top width (m)	Bottom width (m)	Length (m)	Qty. (cum.)	Qty. (kg)	Rate (INR)	Amount (INR)
OGPC	20	14	14	1000	280	0	267	74,760
WBM Grade-3	75	14	14	1000	1050	0	2180	2,289,000
WBM (CBR > 100%)	150	14	14	1000	2100	0	2180	4,578,000
Granular sub-base (CBR > 20%)	175	14.98	15.68	1000	2683	0	2039	5,470,127
Modified subgrade	0	15.68	15.68	1000	0	0	3627	-
Subgrade	500	15.68	17.68	1000	8340	0	393	3,277,620
Embankment	500	17.68	19.68	1000	9340	0	393	3,670,620
Total	1420	-	-	-	-	-	-	-
Guar gum	-	-	-	-	643,552		55	35,395,360
Total								54,755,487
Total in Cr.								5.48

**Table 6 polymers-16-02202-t006:** Pavement design and cost analysis for guar gum in subgrade soil (T9 category).

Case 3: 2% Guar Gum in Only Subgrade Soil								
Length of road (m)		1000		MDD (kg/m^3^)		1820		
Carriageway width (m)		14		Biopolymer		Guar gum		
Effective CBR (%)		2.90		Layer to be used		Subgrade		
Subgrade class		S2		Dosage		2%		
Layer	Thickness (mm)	Top width (m)	Bottom width (m)	Length (m)	Qty. (cum.)	Qty. (kg)	Rate (INR)	Amount (INR)
OGPC	20	14	14	1000	280	0	267	74,760
Bituminous macadam	50	14	14	1000	700	0	7644	5,350,800
WBM (CBR > 100%)	225	14	14	1000	3150	0	2180	6,867,000
Granular sub-base (CBR > 20%)	200	15.18	15.98	1000	3116	0	2039	6,353,524
Modified subgrade	100	15.98	16.38	1000	1618	0	3627	5,868,486
Subgrade	400	15.98	17.58	1000	6712	0	393	2,637,816
Embankment	500	17.58	19.58	1000	9290	0	393	3,650,970
Total	1495	-	-	-	-	-	-	-
Guar gum	-	-	-	-	244,317		55	13,437,424
Total								44,240,780
Total in Cr.								4.42

**Table 7 polymers-16-02202-t007:** Pavement design and cost analysis for guar gum in subgrade and embankment soil (T9 category).

Case 4: 2% Guar Gum in Subgrade and Embankment Soil								
Length of road (m)		1000		MDD (kg/m^3^)		1820		
Carriageway width (m)		14		Biopolymer		Guar gum		
Effective CBR (%)		12.57		Layer to be used		Subgrade and embankment		
Subgrade class		S5		Dosage		2%		
Layer	Thickness (mm)	Top width (m)	Bottom width (m)	Length (m)	Qty. (cum.)	Qty. (kg)	Rate (INR)	Amount (INR)
OGPC	20	14	14	1000	280	0	267	74,760
Bituminous macadam	50	14	14	1000	700	0	7644	5,350,800
WBM (CBR > 100%)	225	14	14	1000	3150	0	2180	6,867,000
Granular sub-base (CBR > 20%)	125	15.18	15.68	1000	1929	0	2039	3,932,721
Modified subgrade	0	15.68	15.68	1000	0	0	3627	-
Subgrade	500	15.68	17.68	1000	8340	0	393	3,277,620
Embankment	500	17.68	19.68	1000	9340	0	393	3,670,620
Total	1420	-	-	-	-	-	-	-
Guar gum	-	-	-	-	643,552		55	35,395,360
Total								58,568,881
Total in Cr.								5.86

## Data Availability

The datasets generated, collected, and analyzed during the current study are available from the corresponding author on reasonable request.

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
