# Peer review of "Effect of Guar Gum Content on the Mechanical Properties of Laterite Soil for Subgrade Soil Application"

_polymers, 2024, doi:10.3390/polym16152202_

Round 1

Reviewer 1 Report

Comments and Suggestions for Authors

The literature review is not written in a proper way to convey the state of the art and put the research in context.  Thus, the merit of the research performed by the authors could not be properly evaluated.  

Comments on the Quality of English Language

Difficult to follow.  Major editing needed.

Author Response

Comment 01: The literature review is not written in a proper way to convey the state of the art and put the research in context.  Thus, the merit of the research performed by the authors could not be properly evaluated.  

Reply: We are thankful to the reviewer for the comments. In the updated manuscript, we have rewritten the literature section to improve clarity and coherence. The updated portions are highlighted in yellow for your convenience.

Reviewer 2 Report

Comments and Suggestions for Authors

The manuscript discusses the application of guar gum biopolymer for laterite soil stabilization, comparing it favourably to traditional methods in terms of mechanical performance. The study clearly articulates its objective of evaluating the physical properties of guar gum-treated soil mixes, focusing on strength enhancement and moisture content optimization. However, it lacks novelty and needs to improve its suitability to be considered for publication. The authors should address the following comments:

1. The abstract lacks scientific quantitative data on the findings which should be added instead of very general descriptive statements, e.g.:

1.1. Line 17-18: the authors discussed the enhancement in the strength and CBR, what are the values? Quantitative data should be included.

1.2: Line 22-23: "...a slightly higher financial investment..." - clarify quantitatively --> how much higher?

2. The introduction section does not provide the readers with an introduction to the study of this paper (e.g.: literature on Guar Gum). Most of these are mentioned in section 1.2. Suggest merging into one section for introduction.

3. Most of the literature discussed in sections 1.1 and 1.3 is irrelevant to the topic discussed in this paper. The introduction must be revised to be concise - simplify to one short paragraph in the Introduction section. 

4. Justify what the mix method is and how they are done. Provide relevant references.

5. Line 117: Specify clearly what are the index properties.

6. The difference between the guar gum used in this paper with the past literature has to be clearly discussed. What is the novelty of this paper compared to the following published papers:

Sujatha, E. R., & Saisree, S. (2019). Geotechnical behaviour of guar gum-treated soil. Soils and Foundations59(6), 2155-2166.

7. Figure numberings are not in order. The qualities are poor, this has to be corrected (i.e. decimal place for Figure 2, background color should be 'clear' instead of grey; Font size and type for Figure 3 axes title should be made bigger; these has to be standardised across all figures).

8. The definition of Gear Gum should not be discussed in Section 2.2 with no references cited. Suggest to include in Introduction.

9. XRD was done on laterite soil, but there are 4 types of soils used in the testing. The XRD results for all types of laterite soil used in the study should be included. Similar to their properties test results. 

10. The discussion on the microstructural study in Section 3.5 is only on LS+2% GG. This is incomplete. Is this the suggested optimal? However, it seems that from the overall observations, LS+3% GG too, demonstrates better performance. Isn't this should be recommended as optimum content? Justify this.

11. The conclusion summarised the internal friction results, however, there is no triaxial test conducted in the study. How were the results obtained?

12. The discussion lacks scientific depth in explaining the results. More precise emphasis should be made on the discussion, their practical implication and applicability. Recommendations should also be provided for future consideration. 

Comments on the Quality of English Language

There were some serious flaws in the writing and grammatical errors, more specifically:

1. Line 11: Vocabulary used seems incorrect: "preferable"? Should it be "favourable"?

2. Line 19: grammatical error: stabilizing Guar gum is different from stabilizing the laterite soil "using" guar gum. Please proofread and revise your statement. 

Author Response

Reviewer #2: Below are some comments that are intended to improve the manuscript.

The manuscript discusses the application of guar gum biopolymer for laterite soil stabilization, comparing it favorably to traditional methods in terms of mechanical performance. The study clearly articulates its objective of evaluating the physical properties of guar gum-treated soil mixes, focusing on strength enhancement and moisture content optimization. However, it lacks novelty and needs to improve its suitability to be considered for publication. The authors should address the following comments:

Comment 01: The abstract lacks scientific quantitative data on the findings which should be added instead of very general descriptive statements, e.g.:

Reply: We are thankful to the reviewer for the comments. Yes, the changes made in the abstract section are highlighted in yellow for your review.

Laterite soil with 2% guar gum show un-soaked CBR increased by 148% and soaked CBR by 192.36%. Cohesiveness and internal friction angle increased by 93.33% and 31.52%, respectively.

Using guar gum in T8 subgrade soil required a 1395 mm pavement depth and cost Rs. 3.83 Crores, 1.52 times more than laterite soil. For T9 subgrade soil, the depth was 1495 mm, costing Rs. 4.42 Crores, 1.72 times more than laterite soil.

Comment 02: Line 17-18: the authors discussed the enhancement in the strength and CBR, what are the values? Quantitative data should be included.

Reply: We are thankful to the reviewer for the comments. Yes, the changes made in the abstract section are highlighted in yellow for your review.

Laterite soil with 2% guar gum show un-soaked CBR increased by 148% and soaked CBR by 192.36%. Cohesiveness and internal friction angle increased by 93.33% and 31.52%, respectively.

Comment 03: Line 22-23: "...a slightly higher financial investment..." - clarify quantitatively --> how much higher?

Reply: We are thankful to the reviewer for the comments. Yes, the changes made in the abstract section are highlighted in yellow for your review.

Using guar gum in T8 subgrade soil required a 1395 mm pavement depth and cost Rs. 3.83 Crores, 1.52 times more than laterite soil. For T9 subgrade soil, the depth was 1495 mm, costing Rs. 4.42 Crores, 1.72 times more than laterite soil.

Comment 04: The introduction section does not provide the readers with an introduction to the study of this paper (e.g.: literature on Guar Gum). Most of these are mentioned in section 1.2. Suggest merging into one section for introduction.

Reply: We are thankful to the reviewer for the comments. Yes, the changes made in the introduction section. Merged in one section.

Comment 05: Most of the literature discussed in sections 1.1 and 1.3 is irrelevant to the topic discussed in this paper. The introduction must be revised to be concise - simplify to one short paragraph in the Introduction section. 

Reply: We are thankful to the reviewer for the comments. Yes, the changes made in the introduction section. Merged in one section.

Comment 06: Justify what the mix method is and how they are done. Provide relevant references.

Reply: We are thankful to the reviewer for the comments. Yes, it is mention in section 2.3.

The powdered biopolymer is mixed with water to make a viscous gel at concentrations of 1%, 2%, and 3% based on soil weight (Wet mixing method).

Comment 07: Line 117: Specify clearly what are the index properties.

Reply: We are thankful to the reviewer for the comments. Yes, it was incorrectly written as index properties. It should only be the engineering properties of the soil. It has been removed from the manuscript.

Comment 08: The difference between the guar gum used in this paper with the past literature has to be clearly discussed. What is the novelty of this paper compared to the following published papers: Sujatha, E. R., & Saisree, S. (2019). Geotechnical behaviour of guar gum-treated soil. Soils and Foundations59(6), 2155-2166.

Reply: We are thankful to the reviewer for the comments. Thank you for your observation. Yes, there is a difference between the geotechnical behavior discussed in the paper on guar gum-treated soil and this manuscript. The published paper utilized a different type of soil, whereas in the current study, we used laterite soil collected from Chiplun, Maharashtra, India. Additionally, in this manuscript, we conducted XRD analysis for a detailed microstructural examination.

Comment 09: Figure numberings are not in order. The qualities are poor, this has to be corrected (i.e. decimal place for Figure 2, background color should be 'clear' instead of grey; Font size and type for Figure 3 axes title should be made bigger; these has to be standardized across all figures).

Reply: We are thankful to the reviewer for the comments. Yes, the changes made in the manuscript. In figure 3, The graph displays Intensity versus 2Ï´ as generated by the software.

Comment 10: The definition of Gear Gum should not be discussed in Section 2.2 with no references cited. Suggest to include in Introduction.

Reply: We are thankful to the reviewer for the comments. Yes, the changes made in the manuscript (guar gum stabilization section) and highlighted in yellow for your review.  

Comment 11: XRD was done on laterite soil, but there are 4 types of soils used in the testing. The XRD results for all types of laterite soil used in the study should be included. Similar to their properties test results.

Reply: We are thankful to the reviewer for the comments. The XRD results and properties of the laterite soil are almost identical because the samples were collected within a 0.5 km radius. This proximity resulted in negligible variation, which is why detailed distinctions were not included in the manuscript.

Comment 12: The discussion on the microstructural study in Section 3.5 is only on LS+2% GG. This is incomplete. Is this the suggested optimal? However, it seems that from the overall observations, LS+3% GG too, demonstrates better performance. Isn't this should be recommended as optimum content? Justify this.

Reply: We are thankful to the reviewer for the comments. Thank you for your observation. In the present study, the optimum dose was determined to be LS + 2% GG, which is why it is discussed in section 3.5. It was mistakenly written as LS + 3% GG in the conclusion section. The error has been corrected.

Comment 13: The conclusion summarized the internal friction results, however, there is no triaxial test conducted in the study. How were the results obtained?

Reply: We are thankful to the reviewer for the comments. The internal friction results mentioned in the conclusion section are indeed derived from the direct shear test results, as explained in detail in section 3.3.

Comment 14: The discussion lacks scientific depth in explaining the results. More precise emphasis should be made on the discussion, their practical implication and applicability. Recommendations should also be provided for future consideration. 

Reply: We are thankful to the reviewer for the comments. We have revised the manuscript to improve the precision of our results presentation. The changes have been implemented accordingly in the updated version.

Comment 15: Line 11: Vocabulary used seems incorrect: "preferable"? Should it be "favourable"?

Reply: We are thankful to the reviewer for the comments. We have incorporated the term "favorable" as suggested. Many researchers have reported that biopolymers enhance soil properties, which has been reflected in the revised abstract section.

Comment 16: Line 19: grammatical error: stabilizing Guar gum is different from stabilizing the laterite soil "using" guar gum. Please proofread and revise your statement. 

Reply: We are thankful to the reviewer for the comments. Yes, the changes made in the abstract section and highlighted in yellow for your review.  

The results shows that using guar gum greatly improves the strength of laterite soil, offering a more environmentally friendly and sustainable alternative to traditional soil additives.

Reviewer 3 Report

Comments and Suggestions for Authors

The research article by Banne et al. describes the effect of guar gum, a biopolymer, on the mechanical properties of Iron rich Laterite Soil for subgrade soil application. The following observations should be clarified before publishing this article.

1.   The tile of the article is very poor and shows very poor writing style. For example the authors have used apostrophe mark in “Laterite soil’s” and the word Biopolymer which must be avoided it can be re-written as “Effect of  Guar Gum Content on the Mechanical Properties of Laterite Soil for Subgrade Soil Application” OR “Effect of  Biopolymer Content on the Mechanical Properties of Laterite Soil for Subgrade Soil Application”.

2.   The abstract do not include the findings of the study which must be included with numbers that percent improve etc.

3.   It is believed that the literature review in the introduction section is not sufficient for providing support to the novelty of the research. The English must be improved in the article to enhance the readability of the text.

4.   Table 1 must give separate properties of type of clay i.e. A, B. C and D.

5.   The figures must follow ascending numbers rather starting from zero again. The caption of Figure 0 is a typo mistake which must be Figure 3. Also the same figure doesn’t mention the type of the clay i.e. A or B or C or D. It must be clarified or must be clearly mentioned in text that all the type of collected clay has same composition.

6.   The authors always write “Guar Gum Biopolymer’ (e.g. line 190, 195 and 197) while it is well documented that guar gum is a biopolymer so it is better to write only guar gum.

7.   In lines 190-196, the detail provided for guar gum has no literature citation while the details provided are not experimentally determined by the authors.

8.   I don’t know why authors have performed XRD of guar gum and determined the elemental composition which has no contribution in the scientific understand of the guar gum interaction with laterite soil. It is strongly recommended to perform FTIR analysis of the guar gum, laterite soil and their mixture to understand the interactions of guar gum and laterite soil for improved mechanical properties.

9.   In sample preparation, lines 219-220, the authors write “when you thoroughly combine …” shows very poor writing style. The authors must also mention the total water content added in the mixture. It seems very difficult to repeat the experiment after reading this sample preparation.

10. Figure 7 (line 231) has been plotted as a smooth line? Is it modeling of the data points? If yes not explained in the text or if not then better to plot data points rather than a straight line because the authors have only three data points.

11. In line 227, authors write that “Guar gum’s hydrogen bonding increases hydrogel formation…” but they did not show any prove of hydrogen bonding. The authors should use FTIR to prove their claim.

12.The authors wrote reference seven in capital. I don’t understand why a different format? In line 253 the spelling of guidelines is incorrect and should be corrected for the type mistake.

13.The authors should add summary before writing conclusion. The whole text and conclusion must be re-write to improve grammar and readability.

Comments on the Quality of English Language

English language requires significant improvement. 

Author Response

Reviewer #3: Below are some comments that are intended to improve the manuscript.

The research article by Banne et al. describes the effect of guar gum, a biopolymer, on the mechanical properties of Iron rich Laterite Soil for subgrade soil application. The following observations should be clarified before publishing this article.

Reply: We are thankful to the reviewer for the comments.

Comment 01: The title of the article is very poor and shows very poor writing style. For example, the authors have used apostrophe mark in “Laterite soil’s” and the word Biopolymer which must be avoided it can be re-written as “Effect of Guar Gum Content on the Mechanical Properties of Laterite Soil for Subgrade Soil Application” OR “Effect of Biopolymer Content on the Mechanical Properties of Laterite Soil for Subgrade Soil Application”.

Reply: We are thankful to the reviewer for the comments. Yes, the title was changed in the updated manuscript.

Comment 02:  The abstract do not include the findings of the study which must be included with numbers that percent improve etc.

Reply: We are thankful to the reviewer for the comments. Yes, the changes made in the abstract section are highlighted in yellow for your review.

Laterite soil with 2% guar gum show un-soaked CBR increased by 148% and soaked CBR by 192.36%. Cohesiveness and internal friction angle increased by 93.33% and 31.52%, respectively.

Using guar gum in T8 subgrade soil required a 1395 mm pavement depth and cost Rs. 3.83 Crores, 1.52 times more than laterite soil. For T9 subgrade soil, the depth was 1495 mm, costing Rs. 4.42 Crores, 1.72 times more than laterite soil.

Comment 03: It is believed that the literature review in the introduction section is not sufficient for providing support to the novelty of the research. The English must be improved in the article to enhance the readability of the text.

Reply: We are thankful to the reviewer for the comments. Yes, the changes made in the introduction section. Merged in one section and We tried improvement in English.

Comment 04: Table 1 must give separate properties of type of clay i.e. A, B. C and D.

Reply: We are thankful to the reviewer for the comments. The properties of the laterite soil are almost identical because the samples were collected within a 0.5 km radius. This proximity resulted in negligible variation, which is why detailed distinctions were not included in the manuscript.

Comment 05: The figures must follow ascending numbers rather starting from zero again. The caption of Figure 0 is a typo mistake which must be Figure 3. Also the same figure doesn’t mention the type of the clay i.e. A or B or C or D. It must be clarified or must be clearly mentioned in text that all the type of collected clay has same composition.

Reply: We are thankful to the reviewer for the comments. Yes, the figure numbering has been verified and The properties of the laterite soil are almost identical [Line 160-161].

Comment 06: The authors always write “Guar Gum Biopolymer’ (e.g. line 190, 195 and 197) while it is well documented that guar gum is a biopolymer so it is better to write only guar gum.

Reply: We are thankful to the reviewer for the comments. Yes, the changes are made in the manuscript.

Comment 07:  In lines 190-196, the detail provided for guar gum has no literature citation while the details provided are not experimentally determined by the authors.

Reply: We are thankful to the reviewer for the comments. Thank you for noting that the citation has been included in the updated manuscript, which shown in yellow colour.

An all-natural biopolymer extracted from the guar seed's endosperm polysaccharide, guar gum is also known by its scientific name, Cyamopsistetragonolobus. Guar gum comes from the guar seed, which is a leguminosae plant [42].

Comment 08: I don’t know why authors have performed XRD of guar gum and determined the elemental composition which has no contribution in the scientific understand of the guar gum interaction with laterite soil. It is strongly recommended to perform FTIR analysis of the guar gum, laterite soil and their mixture to understand the interactions of guar gum and laterite soil for improved mechanical properties.

Reply: We are thankful to the reviewer for the comments. XRD tests conducted on laterite soil stabilized with guar gum provides comprehensive insights into the mineralogical and structural changes induced by the stabilization process and specifically the elemental compositions on engineering properties of laterite soil and guar gum mixes.

Comment 09:  In sample preparation, lines 219-220, the authors write “when you thoroughly combine …” shows very poor writing style. The authors must also mention the total water content added in the mixture. It seems very difficult to repeat the experiment after reading this sample preparation.

Reply: We are thankful to the reviewer for the comments. Yes, the changes are made in the manuscript.

When the laterite soil and gel are mixed, the guar gum evenly coats each soil particle.

The powdered biopolymer is mixed with required amount of water content to make a viscous gel at concentrations of 1%, 2%, and 3% based on soil weight (Wet mixing method).

Comment 10: Figure 7 (line 231) has been plotted as a smooth line? Is it modeling of the data points? If yes not explained in the text or if not then better to plot data points rather than a straight line because the authors have only three data points.

Reply: We are thankful to the reviewer for the comments. Figure 7 illustrates the trend of consistency limits (liquid limit, plastic limit, and shrinkage limit) increasing with the dosage of guar gum. The smooth line represents this increasing trend effectively. It’s not the data points.

Comment 11: In line 227, authors write that “Guar gum’s hydrogen bonding increases hydrogel formation…” but they did not show any prove of hydrogen bonding. The authors should use FTIR to prove their claim.

Reply: We are thankful to the reviewer for the comments. Figure 13 illustrates micrographs depicting the hydrogen bonding between soil particles and guar gum. As highlighted in our XRD analysis, the significant oxygen content of 42.60% in guar gum plays a crucial role in forming robust hydrogen bonds with water molecules, contributing to its effectiveness in soil stabilization.

Comment 12: The authors wrote reference seven in capital. I don’t understand why a different format? In line 253 the spelling of guidelines is incorrect and should be corrected for the type mistake.

Reply: We are thankful to the reviewer for the comments. The changes have been incorporated in the updated manuscript. Thank you.

Comment 13: The authors should add summary before writing conclusion. The whole text and conclusion must be re-write to improve grammar and readability.

Reply: We are thankful to the reviewer for the comments. We have revised the manuscript to improve the precision of our results presentation, conclusions and English. The changes have been implemented accordingly in the updated version.

Comment 14: English language requires significant improvement. 

Reply: We are thankful to the reviewer for the comments. We have revised the manuscript to improve the precision of our results presentation and English. The changes have been implemented accordingly in the updated version.

Round 2

Reviewer 2 Report

Comments and Suggestions for Authors

The manuscript has been revised and can be accepted for publication. 

Author Response

Comment 01: The manuscript has been revised and can be accepted for publication. 

Reply: We are thankful to the reviewer for the comments. Thank you so much for accepting the manuscript.

Reviewer 3 Report

Comments and Suggestions for Authors

The authors have submitted a revised version of their manuscript, but it requires additional revisions before it can be accepted for publication in the Polymers Journal. No significant improvment observed in the revised version.

1.      The quality of English language is not up to the standard

2.      The authors fail to highlight the novelty and significance of their research in both the abstract and introduction section, making it difficult for readers to understand the contribution of this study to the field. Please revise the abstract and introduction to clearly articulate the innovative aspects and impact of this work.

3.      The introduction section suffers from poor connectivity between sentences and ideas, making it difficult to follow. Additionally, the section lacks sufficient references to support the statements made. Please revise the introduction to improve the coherence and add relevant references to strengthen the foundation of the manuscript.

4.      The way of writing the references are also incorrect. For example the authors wrote [6], [7], [8]. It should be like [6-8].

5.      Line 160 again spelling mistake locations rather than loactions. The authors should proof read the manuscript before submitting it to the journal

6.      At almost 10-12 places again the authors wrote Guar Gum Biopolymers.

7.      The authors failed to address comment 8 regarding the FTIR in their revised manuscript. Please provide a response to this comment and include the necessary revisions to the FTIR analysis section

8.      In the sample preparation section, the authors mention mixing the powdered biopolymer with water to form a viscous gel at concentrations of 1%, 2%, and 3% based on soil weight. Could you please clarify what these percentage values represent (e.g., weight/weight, weight/volume)? Additionally, could you express these concentrations in terms of milliliters (mL) for better understanding??

9.      Figure 7 still needs further explanation

10.  There is some discrepancy in the Oxygen content in the author response and in the XRD report.

11.  Can you tell whether XRD is an optimal tool for illustrating hydrogen bonding? Could you please reassess and advise on the most suitable technique for visualizing this phenomenon

12.  The format for reference 43 and 44 is incorrect.

Comments on the Quality of English Language

The quality of english language used is still not up to the mark. Many typos were still in the manuscript. 

Author Response

Reviewer #3: Below are some comments that are intended to improve the manuscript.

The authors have submitted a revised version of their manuscript, but it requires additional revisions before it can be accepted for publication in the Polymers Journal. No significant improvement observed in the revised version.

Reply: We are thankful to the reviewer for the comments.

Comment 01: The quality of English language is not up to the standard.

Reply: We are thankful to the reviewer for the comments. We have made the necessary changes to improve the quality of English in the manuscript, ensuring it meets the required standards. The changes are highlighted in yellow throughout the document.

Comment 02:  The authors fail to highlight the novelty and significance of their research in both the abstract and introduction section, making it difficult for readers to understand the contribution of this study to the field. Please revise the abstract and introduction to clearly articulate the innovative aspects and impact of this work.

Reply: We are thankful to the reviewer for the comments. We have incorporated the novelty and significance of the present study into the manuscript. The present study shows the effect of guar gum on laterite soil. For your reference, please see lines 25-28 in the abstract section and lines 105-112 in the introduction section. The changes are highlighted in yellow color.

Comment 03: The introduction section suffers from poor connectivity between sentences and ideas, making it difficult to follow. Additionally, the section lacks sufficient references to support the statements made. Please revise the introduction to improve the coherence and add relevant references to strengthen the foundation of the manuscript.

Reply: We are thankful to the reviewer for the comments. We have made changes to the introduction section to improve coherence as suggested. The introduction has also been reduced in length and the manuscript has been revised accordingly to address your comments [Please refer section 1 Introduction].

Comment 04: The way of writing the references are also incorrect. For example the authors wrote [6], [7], [8]. It should be like [6-8].

Reply: We are thankful to the reviewer for the comments. Yes, the changes are made in the manuscript [Please refer line 55].

Comment 05: Line 160 again spelling mistake locations rather than loactions. The authors should proof read the manuscript before submitting it to the journal.

Reply: We are thankful to the reviewer for the comments. Yes, the changes are made in the manuscript [Please refer line 127].

Comment 06: At almost 10-12 places again the authors wrote Guar Gum Biopolymers.

Reply: We are thankful to the reviewer for the comments. Yes, the changes are made in the manuscript.

Comment 07:  The authors failed to address comment 8 regarding the FTIR in their revised manuscript. Please provide a response to this comment and include the necessary revisions to the FTIR analysis section.

Reply: We are thankful to the reviewer for the comments. Respected reviewer, In the present study, we did not conduct FTIR analysis. Instead, we relied on XRD analysis based on the literature to gain insights into the material composition and structural changes. XRD tests conducted on laterite soil stabilized with guar gum provides comprehensive insights into the mineralogical and structural changes induced by the stabilization process and specifically the elemental compositions on engineering properties of laterite soil and guar gum mixes. Additionally, SEM analysis was performed to observe the bonding between laterite soil particles and guar gum.

Comment 08: In the sample preparation section, the authors mention mixing the powdered biopolymer with water to form a viscous gel at concentrations of 1%, 2%, and 3% based on soil weight. Could you please clarify what these percentage values represent (e.g., weight/weight, weight/volume)? Additionally, could you express these concentrations in terms of milliliters (mL) for better understanding??

Reply: We are thankful to the reviewer for the comments. The percentage values mentioned in the sample preparation section represent weight/weight (w/w) concentrations. Specifically, they indicate the ratio of the weight of the powdered guar gum to the weight of the laterite soil. For example, a 1% concentration means that 10 gram of the guar gum was mixed with 1000 grams of laterite soil to form the viscous gel. Similarly, 2% and 3%. [Please Refer Line 173-178].

Comment 09:  Figure 7 still needs further explanation.

 Reply: We are thankful to the reviewer for the comments. Yes, we have added the requested explanation. Please refer to Section 3.1, highlighted in yellow, for the detailed addition.

Comment 10: There is some discrepancy in the Oxygen content in the author response and in the XRD report.

Reply: We are thankful to the reviewer for the comments. We have included the oxygen content data in the manuscript. Specifically, the oxygen content in the laterite soil was measured at 44.30%, in the guar gum at 42.60%, and in the combination of laterite soil + 2% guar gum at 48.50%. Please refer to the following sections for detailed information: lines 139-144 for laterite soil, lines 157-167 for guar gum, and lines 254-269 for the laterite soil and guar gum mix.

Comment 11: Can you tell whether XRD is an optimal tool for illustrating hydrogen bonding? Could you please reassess and advise on the most suitable technique for visualizing this phenomenon?

Reply: We are thankful to the reviewer for the comments. XRD can provide valuable information about the crystalline structure of materials and its elemental composition, it is not the most optimal tool for illustrating hydrogen bonding specifically. We acknowledge this limitation.

But, Laterite Soil and guar gum mix, with a high oxygen content, such as 48.50%, typically have a significant presence of oxygen-containing compounds like silicates, alumino-silicates, and various oxides. Oxygen atoms in these compounds can act as hydrogen bond acceptors, contributing to the formation of hydrogen bonds with hydrogen donors, such as water molecules or organic matter present in the soil. The high oxygen content suggests a strong potential for hydrogen bonding, which can influence the soil's physical and chemical properties. These hydrogen bonds enhance the soil's ability to retain moisture, impact its plasticity, and contribute to the stability of organic and inorganic complexes within the soil matrix. In practical terms, this means that the soil may exhibit improved cohesion, potentially higher plastic and liquid limits, and a greater ability to maintain structure. [Please refer section 3.5.1, Line – 256-262].

Comment 12: The format for reference 43 and 44 is incorrect.

Reply: We are thankful to the reviewer for the comments. We have made the necessary changes in the manuscript. References number 43 and 44 have been updated to correctly cite the Indian Standard codes.

Round 3

Reviewer 3 Report

Comments and Suggestions for Authors

The article can be accepted in its present form; however, the authors may wish to consider the following points for further improvement

1. Reference can be written in a single bracket. For example the author wrote [20], [22], [24]. It can be written as [20, 22, 24]. Please see this throughout the manuscript

2. Remove grey background from Figure 8, 9, 10

Comments on the Quality of English Language

satisfactory.